# FLAIRBrainSeg: Fine-grained brain segmentation using FLAIR MRI only

**Edern Le Bot**[1]                                                    EDERN.LE-BOT@U-BORDEAUX.FR
[1] *Univ. Bordeaux, CNRS, Bordeaux INP, LaBRI, UMR 5800, F-33400 Talence, France*

**Rémi Giraud** [2]                                                   REMI.GIRAUD@BORDEAUX-INP.FR
[2] *Univ. Bordeaux, Bordeaux INP, IMS, CNRS, UMR 5218, F-33400 Talence, France*

**Boris Mansencal**[1]                                            BORIS.MANSENCAL@U-BORDEAUX.FR
**Thomas Tourdias**[3,4]                                         THOMAS.TOURDIAS@CHU-BORDEAUX.FR
[3] *Inserm U1215 - Neurocentre Magendie, Bordeaux F-33000, France*
[4] *Service de Neuroimagerie diagnostique et thérapeutique, CHU de Bordeaux, F-33000 Bordeaux, France*

**Josè V. Manjon**[5]                                                      JMANJON@FIS.UPV.ES
[5] *Instituto de Aplicaciones de las Tecnologías de la Información y de las Comunicaciones Avanzadas, Universitat Politècnica de València, Camino de Vera s/n, 46022, Valencia, Spain*

**Pierrick Coupé**[1]                                             PIERRICK.COUPE@U-BORDEAUX.FR

**Editors:** Accepted for publication at MIDL 2025

## Abstract

This paper introduces FLAIRBrainSeg, a novel method for fine-grained segmentation of brain structures using only FLAIR MRIs, specifically targeting cases where access to other imaging modalities is limited. By leveraging existing automatic segmentation methods, we train a network to approximate segmentations, typically obtained from T1-weighted MRIs. Our method produces segmentations of 132 structures and is robust to multiple sclerosis lesions. Experiments on both in-domain and out-of-domain datasets demonstrate that our method outperforms modality-agnostic approaches based on image synthesis, the only currently available alternative for performing brain parcellation using FLAIR MRI alone. This technique holds promise for scenarios where T1-weighted MRIs are unavailable or to reduce acquisition time, and offers a valuable alternative for clinicians and researchers in need of reliable anatomical segmentation.

**Keywords:** Fine-grained segmentation, Magnetic Resonance Imaging, FLAIR, Convolutional Neural Networks

## 1. Introduction

**Problem Statement**: Accurate and detailed brain segmentation is essential for various clinical applications, including diagnosis, treatment planning, and monitoring of neurological diseases. Traditionally, T1-weighted (T1w) MRI has been the standard modality for brain segmentation. However, a significant limitation of relying solely on T1w MRIs is that they are not always available, particularly in certain clinical settings. For example, in the management of acute stroke patients, T1w images are not obtained as part of the initial imaging protocol (Powers et al., 2019). Moreover, guidelines for diagnosing Multiple Sclerosis (MS) (Wattjes et al., 2021; Schmierer et al., 2019) consider T1w acquisition optional, recommending only contrast-enhanced T1w images. If lesion segmentation and

brain volumetry can be effectively performed using only FLAIR images, the acquisition of T1w sequences could become redundant (Goodkin et al., 2021), allowing for a reduction in protocol acquisition times.

**Whole brain segmentation based on T1w MRIs:** A large number of currently available segmentation methods rely on T1w MRIs for generating anatomical segmentations. Due to the challenges of applying deep learning (DL) methods in the context of limited training data, many existing methods prioritize coarse-grained segmentation (*e.g.*, fewer than 50 anatomical structures) (Henschel et al., 2020; Roy et al., 2019). In contrast, fine-grained segmentation, which delineates over 100 structures, provides a more detailed understanding of brain anatomy, including cortical parcellation. The increased complexity of fine-grained segmentation, compared to coarse-grained approaches, necessitates the adoption of specialized strategies such as patch-based methods and model ensembling (Coupé et al., 2020; Huo et al., 2019), or architecture improvements (Hatamizadeh et al., 2021) to achieve meaningful and accurate results.

**Modality-agnostic whole brain segmentation:** Modality-agnostic methods, make an attempt at resolving domain-shift differences in contrast and appearances in between different acquisition modalities. SynthSeg (Billot et al., 2023b), achieves this by introducing a domain randomization strategy to randomize the appearance of anatomical structures during training. In SynthSeg$^+$ (Billot et al., 2023a), the authors improve upon this, in order to obtain more detailed segmentations, where the initial segmentation obtained by SynthSeg is refined using a second "denoising" network. A final network is used to perform cortical parcellation, and allows for the obtention of a final fine-grained segmentation, at the cost of using multiple deep-learning networks. It is important to note that unsupervised (although limited to coarse-grained) modality-agnostic segmentation also exists. Notably, SAMSEG (Puonti et al., 2016), uses a Bayesian framework, relying on an intensity clustering algorithm and a probabilistic brain atlas for the production of coarse-grained segmentations.

To address the challenges of brain segmentation in the absence of T1w images, we propose a novel fine-grained brain segmentation based on FLAIR MRI only. We compare the accuracy of a domain-randomization strategy with randomized tissue intensities, as used by SynthSeg, to more traditional and direct "supervised learning" approaches based on labeled FLAIR data. Specifically, we highlight differences in the segmentations obtained in the presence of unexpected anomalies, such as multiple sclerosis lesions, which can significantly alter the appearance of MRI images and challenge the robustness of segmentation methods.

## 2. Material & Methods

### 2.1. Datasets:

- Training dataset: 3577 pairs of T1w and FLAIR MRIs selected after the QC steps introduced below from the OFSEP (Vukusic et al., 2020) database were used for training data. The OFSEP database contains images from 36 different sites (with multiple different scanners) in France, from clinically diagnosed patients with multiple sclerosis, at multiple stages of advancement. T1w and FLAIR images selected from the OFSEP database had a wide range of resolutions ranging from 0.25x0.25x0.5mm to 2x2x3mm. However most available images had a 1mm isotropic sampling.

- Testing dataset:

  A stratified train-test split of all available images (prior to QC) was used to create the testing sets, ensuring balance across age groups and sexes.

  – In-domain testing set: 100 pairs of images, not included in the training set, from the OFSEP database, were used for validation.

  – Out-of-domain testing set: to perform the comparison on out-of-domain images, a selection of 100 pairs of images, from the UKBiobank (Bycroft et al., 2018) database, contains clinically normal patients from 3 different sites in the United Kingdom.

Table 1: Dataset overview. Please note that the number of subjects and dataset statistics are reported *after all the quality control steps.* We include the range of Expanded Disability Status Scale (EDSS) ratings to provide an idea of the stage of progression of MS in the in-domain dataset (i.e., OFSEP).

|  | Dataset | Number of subjects | Average age (years) | Sex distribution | EDSS ratings |
|---|---|---|---|---|---|
| **Train Set** | OFSEP | 3577 | $41.77 \pm 11.56$ | 2613F/964M | $[0, 9.0]$ |
| **Test Set** | OFSEP | 100 | $41.93 \pm 12.09$ | 73F/27M | $[0, 6.0]$ |
|  | UKB | 100 | $64.80 \pm 11.01$ | 50F/50M | n/a |

## 2.2. Preprocessing:

All the used pairs of FLAIR and T1w MRIs from the OFSEP database were processed using the DeepLesionBrain pipeline (Kamraoui et al., 2022). The following steps were applied during the preprocessing : 1) all images were denoised using (Manjón et al., 2010) 2) images were corrected using N4ITK bias-field correction (Tustison et al., 2010) 3) FLAIR images were linearly registered to the corresponding T1w MRI images, using ANTS (Avants et al., 2011) 4) FLAIR and T1w images were transformed to the MNI space. This last step was achieved by first registering the T1w images to the McGill MNI template (Fonov et al., 2011). The resulting T1w-to-MNI transformation was then applied to the FLAIR images to ensure consistent alignment in the MNI space.

## 2.3. T1w-based segmentation:

The segmentation of lesions using FLAIR MRI and fine-grained brain segmentation using T1w were performed in a sequential pipeline. First, DeepLesionBrain (Kamraoui et al., 2022) was employed to segment MS lesions. The goal of producing lesion segmentation was to enable masking of MS lesions in the T1w image, thereby facilitating more accurate anatomical segmentation by AssemblyNet (Coupé et al., 2020). Next, the lesion map was used to perform T1w inpainting (Manjón et al., 2020). Finally, AssemblyNet was applied to the inpainted T1w MRI (see Fig. 1), resulting in a fine-grained

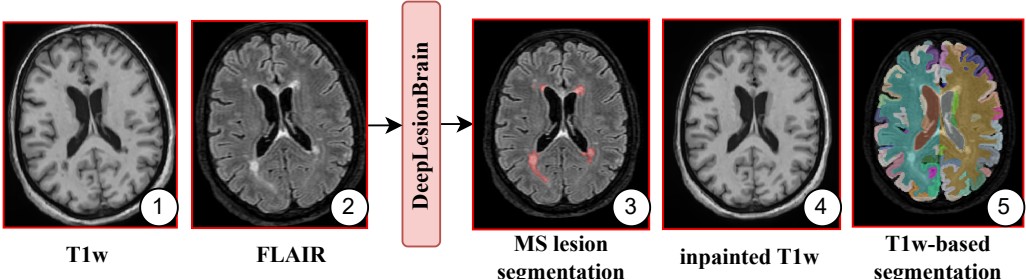

Figure 1: Schematic overview of the T1w-based segmentation construction. From left to right, the images are : the (1) T1w and (2) FLAIR images, the (3) lesion maps, (4) inpainted T1w and (5) the obtained fined-grained segmentation.

segmentation of 132 structures following the Neuromorphometrics protocol (see https://neuromorphometrics.com/). AssemblyNet was selected for its proven test-retest reliability across multiple acquisitions in clinical settings (Coupé et al., 2020), ensuring robustness in varied imaging conditions. Additionally, it was specifically trained for fine-grained segmentation, aligning with the study's objective of achieving detailed anatomical delineation. After verification for possible misalignment between T1w and FLAIR MRIs, the T1w-based segmentation can be used as target labels for FLAIR images.

### 2.4. Quality control

**MNI misalignment:** First, we assessed possible misalignment of the T1w-based segmentation and the FLAIR *training image* by first controlling registration errors between the FLAIR image and the MNI template, using RegQCNet (Denis de Senneville et al., 2020), a robust deep-learning-based method designed to estimate registration errors to the MNI space. In addition to detecting misalignment during registration, RegQCNet facilitated the exclusion of images with excessively poor quality, ensuring a more reliable dataset for analysis.

**Segmentation misalignment:** To assess possible misalignment between the generated segmentation and the FLAIR image, we generated a synthetic FLAIR-like image (see Fig. 2 at left), from the segmentation as follow : 1) For each anatomical structure in the segmentation, we extracted all corresponding voxel intensities from the FLAIR image, then we computed the local median intensity within each region to obtain an estimate of the typical intensity for that structure. 2) We generated the synthetic image by replacing every voxel within a labeled structure with its corresponding median intensity value observed above. 3) To simulate the effect of partial volume averaging, we applied a uniform filter over the synthetic image. Finally, we computed the correlation between the real FLAIR image and the synthetic FLAIR-like image based on T1w segmentation. The computed score reflects the alignment of the segmentation (*i.e.,* T1w MRI) and the FLAIR image, where higher correlation values indicate better spatial alignment. To identify and reject possibly misaligned images, we defined a rejection threshold based on the computed correlation metrics: given $S = \{s_1, s_2, \ldots, s_n\}$, where $s_i$ is the computed correlation scores between the synthetic and the flair image, by fitting a two-component (*i.e.* $K = 2$) Gaussian Mixture

Model (GMM) $p(S) = \sum_{i=1}^{K} \phi_i \mathcal{N}(\mu_i, \Sigma_i)$ on the computed correlation scores, and selecting threshold $= \mu_0 + z^{0.9}\sigma_0$, where $\mu_0$ and $\sigma_0$ are the mean and standard deviation of the lower Gaussian component corresponding to the misaligned FLAIR and segmentation, and $z^{0.9}$ is the 90th percentile of a standard normal distribution. The decision to fit a two-component GMM was based on the hypothesis that the correlation scores for failed and successful registrations would naturally cluster into two distinct distributions. Specifically, successful registrations are expected to yield high correlation scores, forming one Gaussian component, while failed registrations, with lower correlation scores, form a separate component (see Fig. 2 at right).

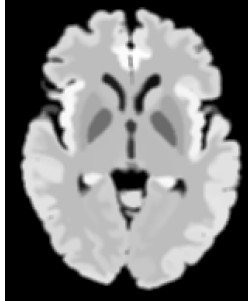 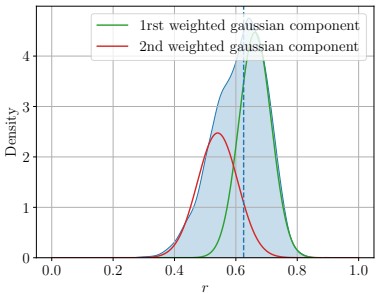

Figure 2: **Left**: Illustration of a synthetic FLAIR-like image created from T1w-based segmentation as part of the quality control process to address registration discrepancies. **Right**: Distribution of correlation scores on the training set. The vertical dotted line corresponds to the selected threshold.

**Visual assessment of testing images**: Lastly, we conducted a human visual inspection of the registration process for all the *testing images*. The authors systematically reviewed the images for potential misalignment, specifically assessing discrepancies between the segmentation and the underlying anatomical structures. Any images exhibiting misalignment or visible defects were excluded from the testing dataset. For the in-domain dataset, 13 images were removed following this visual assessment, while none of the selected test images in the out-of-domain dataset required removal. To replace the removed images, a new test split to select replacement test images from the pool of available images was performed.

## 2.5. Deep learning model description

We used a 3D U-Net derived from the 3D configuration of the nnU-Net (Isensee et al., 2021). The networks were trained using SGD with momentum ($\alpha = 0.99$), with a batch size of 4. A polynomial learning rate (lr) scheduling policy was used, where $lr(t) = (1 - t/t_{max})^{0.9}$, with an initial learning rate of 0.1. Similarly to nnU-Net, a combination of Dice and Cross-Entropy loss was used during training.

At each iteration, batch samples were randomly selected from the training dataset, before applying nnU-Net data augmentations, and randomly sampling a patch of size $112 \times 128 \times 112$ voxels. Augmentations introduced this way included linear spatial transforms (scaling, rotation), intensity transforms (Gaussian noise, Gaussian blur, brightness, contrast, and gamma transforms), and low-resolution resampling (up to 4mm). Unlike the

default nnU-Net configuration, augmentations did not include image flipping, due to neuromorphometrics labels being lateralized. The segmentation network was trained during 1000 epochs, where each epoch was made of 250 iterations.

While alternative methods, such as patch-based approaches (Coupé et al., 2020; Huo et al., 2019; Henschel et al., 2020), are often employed to address the complexities of fine-grained whole-brain segmentation, nnU-Net was selected for its robustness, ease of reproducibility, and established performance across multiple medical imaging segmentation challenges (Isensee et al., 2024). Its adoption as a baseline allows for consistent comparisons while avoiding the variability introduced by more complex, task-specific strategies.

## 2.6. Validation framework

During our experiments, we compared FLAIRBrainSeg with SynthSeg (Billot et al., 2023b) and SynthSeg$^+$ (Billot et al., 2023a). As previously described, SynthSeg relies on a single network to perform coarse-grained segmentation (*i.e.*, 34 structures) while SynthSeg$^+$ uses a cascade of four networks to produce fine-grained segmentation (*i.e.*, 99 structures). To ensure a fair comparison, we performed two experiments.

- **Experiment 1:** We retrained SynthSeg (using the code available at: https://github.com/BBillot/SynthSeg) using our T1w-based segmentations. Therefore, we obtained SynthSeg-132, a version of SynthSeg segmenting 132 structures according to the Neuromorphometrics protocol. Parameters proposed by (Billot et al., 2023b) were re-used for training, except for the number of iterations, where we used an increased number of 500 000 steps instead of 300 000 steps to account for the increased number of structures to segment. For evaluation, since the Neuromorphometrics protocol does not include non-brain tissues, we skull-stripped the test images with the foreground labels of the ground-truth segmentations. This can advantage SynthSeg-132, since other methods might estimate structures outside the brain mask. However, we wanted to make a fair comparison and be as close as possible to the original SynthSeg training, where every part of the image had corresponding labels.

- **Experiment 2:** For SynthSeg$^+$, we used the binary provided by Freesurfer (7.4.1) to segment our testing datasets. To compare SynthSeg$^+$, SynthSeg-132 and FLAIRBrainSeg which follow different protocols, we manually selected the 35 corresponding structures between Freesurfer protocol of SynthSeg$^+$ and Neuromorphometrics protocol of SynthSeg-132 and FLAIRBrainSeg. We emphasize that we acknowledge that this approach is not perfect, as differences may exist not only in the identified structures but also in their borders across protocols.

## 3. Results

The quantitative results based on the average Dice similarity coefficient (DSC) are presented in Fig. 3 and in Tab. 2, alongside the $95^{th}$ percentile of the surface distance (SD95). A comparison of SynthSeg-132 and FLAIRBrainSeg by structure groups is detailed in Appendix B. We assessed the statistical significance of our results by using a Wilcoxon signed-rank test. All $p$-values are inferior to $10^{-13}$ except for in-domain Experiment 2 between SynthSeg-132 and SynthSeg$^+$ ($p = 0.017$). Results from Experiment 1 show that on 132 structures,

FLAIRBrainSeg outperforms SynthSeg-132 method by a substantial margin (respectively 7pp and 10pp higher DSC) on the in-domain and out-of-domain datasets.

Additionally, results from Experiment 2 shows that FLAIRBrainSeg also outperforms SynthSeg-132 and SynthSeg$^+$ on the selected 35 structures used for comparison. SynthSeg-132 shows a statistically significant but minimal improvement over SynthSeg$^+$ (0.5pp higher DSC, $p = 0.017$) on the in-domain testing set; while SynthSeg$^+$ shows a statistically significant improvement in accuracy compared to SynthSeg-132 on the out-of-domain testing set (1pp higher DSC, $p < e^{13}$). Additionally, the variability and spread of accuracy distribution are reduced on the out-of-domain testing set compared to the in-domain set. This is consistent with the experiment design, as this second testing set contains higher quality images from clinically normal patients, less challenging to the segmentation models than the OFSEP clinical data of MS patients used as in-domain.

Table 2: Table report of Mean Dice Score and voxel-wise 95th Surface Distance (SD95) for each testing dataset on Experiment 1

|  |  |  | **Mean** | **Min** | **50%** | **Max** |
|---|---|---|---|---|---|---|
| In-Domain | FLAIRBrainSeg | Dice | $0.90 \pm 0.02$ | 0.75 | 0.91 | 0.93 |
|  |  | SD95 | $0.56 \pm 0.13$ | 0.35 | 0.53 | 1.15 |
|  | SynthSeg | Dice | $0.83 \pm 0.01$ | 0.73 | 0.83 | 0.86 |
|  |  | SD95 | $2.31 \pm 0.86$ | 1.15 | 2.11 | 5.70 |
| Out-of-domain | FLAIRBrainSeg | Dice | $0.91 \pm 0.01$ | 0.86 | 0.91 | 0.93 |
|  |  | SD95 | $0.56 \pm 0.07$ | 0.40 | 0.54 | 0.73 |
|  | SynthSeg | Dice | $0.81 \pm 0.01$ | 0.76 | 0.81 | 0.83 |
|  |  | SD95 | $1.48 \pm 0.20$ | 1.34 | 1.45 | 2.27 |

Examples of segmentation results from the in-domain and out-of-domain test sets are shown at the top of Fig. 4 and in Appendix A. The inaccuracies observed in SynthSeg-132 highlight that its training scheme, designed to induce resilience across multiple contrasts and modalities, may reduce its performance when compared to a modality-specific model. Additionally, due to synthetic training images used by SynthSeg being generated from the segmentation, and since the segmentation does not include a label for the lesion, white matter hyperintensities from MS lesions are never seen in training. As a result, they often end up misclassified (for example, in Fig. 4, MS lesions are incorrectly classified as cortical and ventricles structures in the highlighted area for SynthSeg). While not surprising, this is not the case for FLAIRBrainSeg, as the structures are accurately described in the output segmentation. Therefore, in light of those comparisons, while SynthSeg and SynthSeg$^+$ offer the advantage of adaptability across multiple modalities, specialized models (such as FLAIRBrainSeg) might be preferable when targeting a specific modality.

While SynthSeg and FLAIRBrainSeg both employ single models for their respective segmentation tasks, SynthSeg$^+$ makes use of four different CNN models. This difference in model architecture directly impacts computational resource requirements and runtime. The inclusion of a post-processing step during SynthSeg and SynthSeg$^+$ inference also adds additional complexity, extending the overall runtime compared to FLAIRBrainSeg. For instance, when processed on GPU, the processing of 10 cases (without the inclusion of the

Figure 3: Average DSC for Experiment 1 and Experiment 2. **Top:** results on the in-domain dataset (i.e., OFSEP), **Bottom:** Results on the out-of-domain dataset (i.e., UKB), **Left:** Comparison of SynthSeg-132 and FLAIRBrainSeg over the 132 structures of the Neuromorphometrics protocol (Experiment 1), **Right:** Comparison of SynthSeg-132, SynthSeg$^+$, and FLAIRBrainSeg on the 35 common structures between Freesurfer and Neuromorphometrics protocols (Experiment 2).

**In-domain testing set**

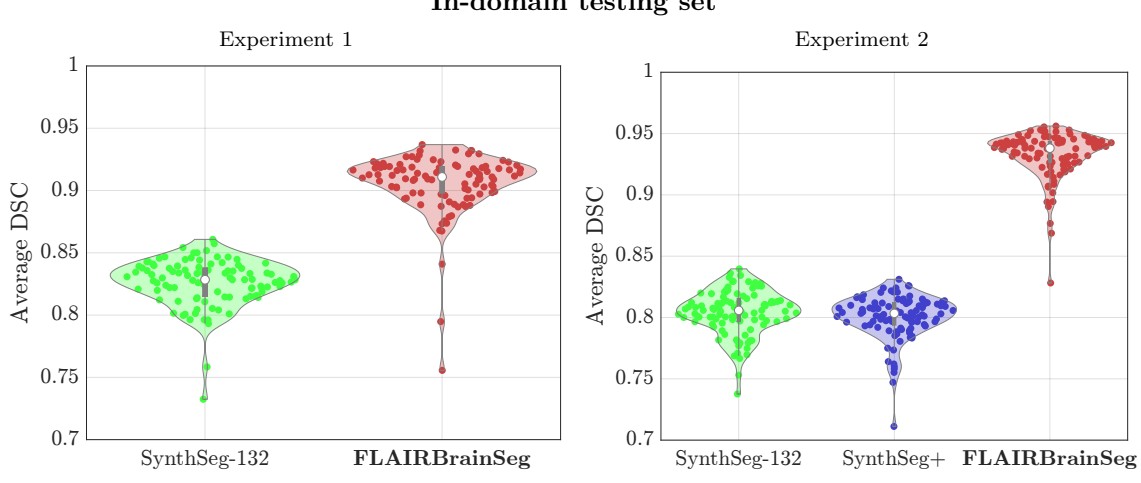

**Out-of-domain testing set**

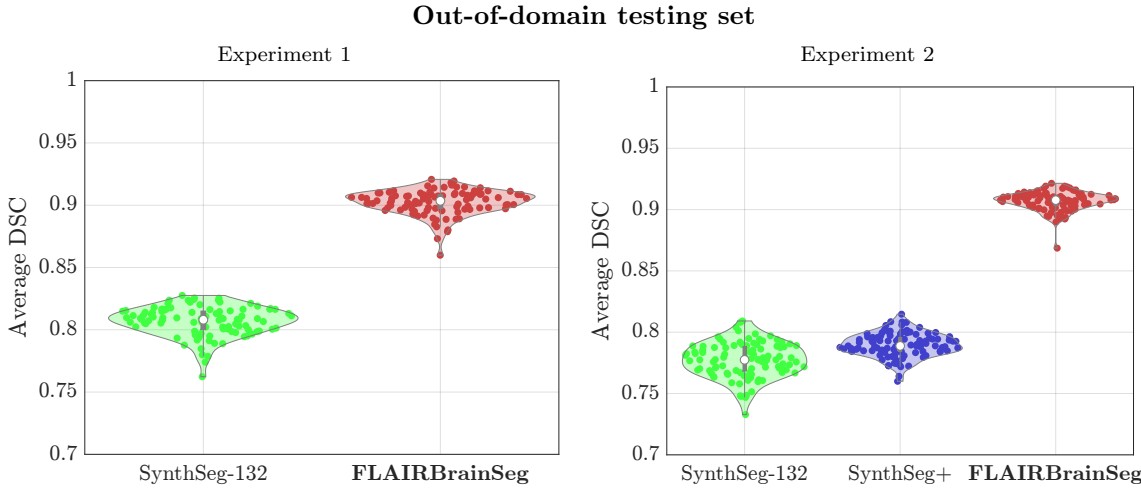

pre-processing time) requires 29.1 seconds for FLAIRBrainSeg, 160.6 seconds for SynthSeg$^+$, and 120.8 seconds for SynthSeg.

## 4. Conclusion

In this work, we introduce FLAIRBrainSeg, a novel framework designed for fine-grained segmentation of FLAIR MRIs. The framework enables the production of detailed anatomical segmentations typically obtained from reliable T1w MRI-based methods, but solely

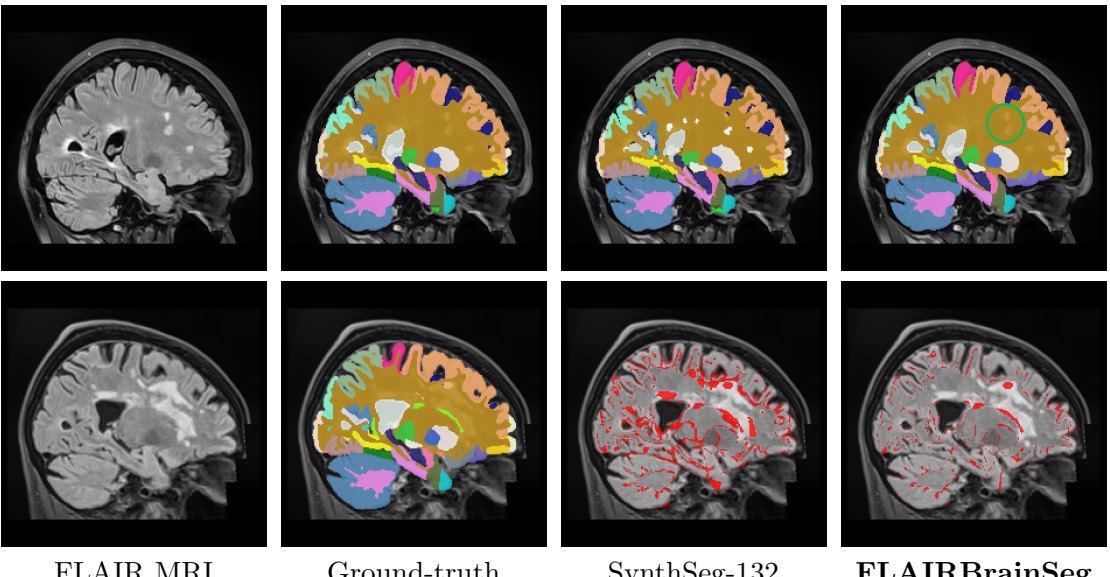

| FLAIR MRI | Ground-truth | SynthSeg-132 | **FLAIRBrainSeg** |

Figure 4: Examples of segmentation of 132 structures obtained with FLAIRBrainSeg and SynthSeg-132 in Experiment 1 are shown on different slices and images, for the in-domain (top) and out-of-domain (bottom) testing sets. The images were selected from cases with median mean DSC scores for SynthSeg-132 to ensure a fair representation of its performance. The top row highlights the accuracy of FLAIRBrainSeg, even in the presence of MS lesions, which are incorrectly labeled as ventricle or cortex by SynthSeg. The bottom row illustrates the differences between the segmentations obtained and the ground-truth labels on another example, further emphasizing the advantages of FLAIRBrainSeg in handling modality-specific challenges.

using FLAIR MRIs. Our results demonstrate the effectiveness of this modality-specific approach, particularly in addressing challenges posed by abnormalities such as white matter hyperintensities. Furthermore, we highlight the limitations of domain randomization techniques, particularly when dealing with pathological conditions in FLAIR MRIs, underlying the need for tailored methods for modality-specific segmentation tasks. A current limitation of our work is the generalization of FLAIRBrainSeg to low-resolution FLAIR images although our method was trained using down-sampling as data augmentation. In clinical practice, FLAIR scans are often acquired at a coarse resolution or as 2D slices with limited coverage along the z-axis. This can affect the image appearance after the linear resampling to the 1 mm template resolution, potentially decreasing segmentation accuracy. Future improvements could leverage recent advancements in super-resolution generative models (Morell-Ortega et al., 2024) to enhance the quality of low-resolution FLAIR images before segmentation. Future work will also focus on enabling the processing of FLAIR images with various types of lesions and disease-related anomalies, in particular incorporating patients with strokes and tumors. This will enhance the applicability of FLAIRBrainSeg and ensure its relevance in clinical settings. In the future, we plan to release this tool for free on https://volbrain.net/.

## Acknowledgments

We would like to express our gratitude to Benjamin Billot for his assistance in setting up the comparison with SynthSeg. His support was instrumental in enhancing the quality of the present work. This research has been conducted using the UK Biobank Resource under application number 80509. Data collection has been supported by a grant provided by the French State and handled by the "Agence Nationale de la Recherche," within the framework of the "Investments for the Future" programme, under the reference ANR-10-COHO-002, Observatoire Français de la Sclérose en Plaques (OFSEP)" and "Eugène Devic EDMUS Foundation against multiple sclerosis." This work was also granted access to the HPC resources of IDRIS under the allocation 2022-AD011013926R1 made by GENCI. Additionally, this work benefited from the support of the project HoliBrain of the French National Research Agency (ANR-23-CE45-0020-01). Moreover, this project is supported by the Precision and global vascular brain health institute funded by the France 2030 investment plan as part of the IHU3 initiative (ANR-23-IAHU-0001). Finally, this study received financial support from the French government in the framework of the University of Bordeaux's France 2030 program / RRI "IMPACT and the PEPR StratifyAging.

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

## Appendix A. Qualitative comparison of segmentation accuracies

Fig. 5 shows visual examples of the obtained segmentations with FLAIRBrainSeg, SynthSeg-132, and SynthSeg$^+$. Note that the SynthSeg$^+$ segmentation includes only the 35 selected structures for comparison, excluding cortical structures and cerebellar gray matter.

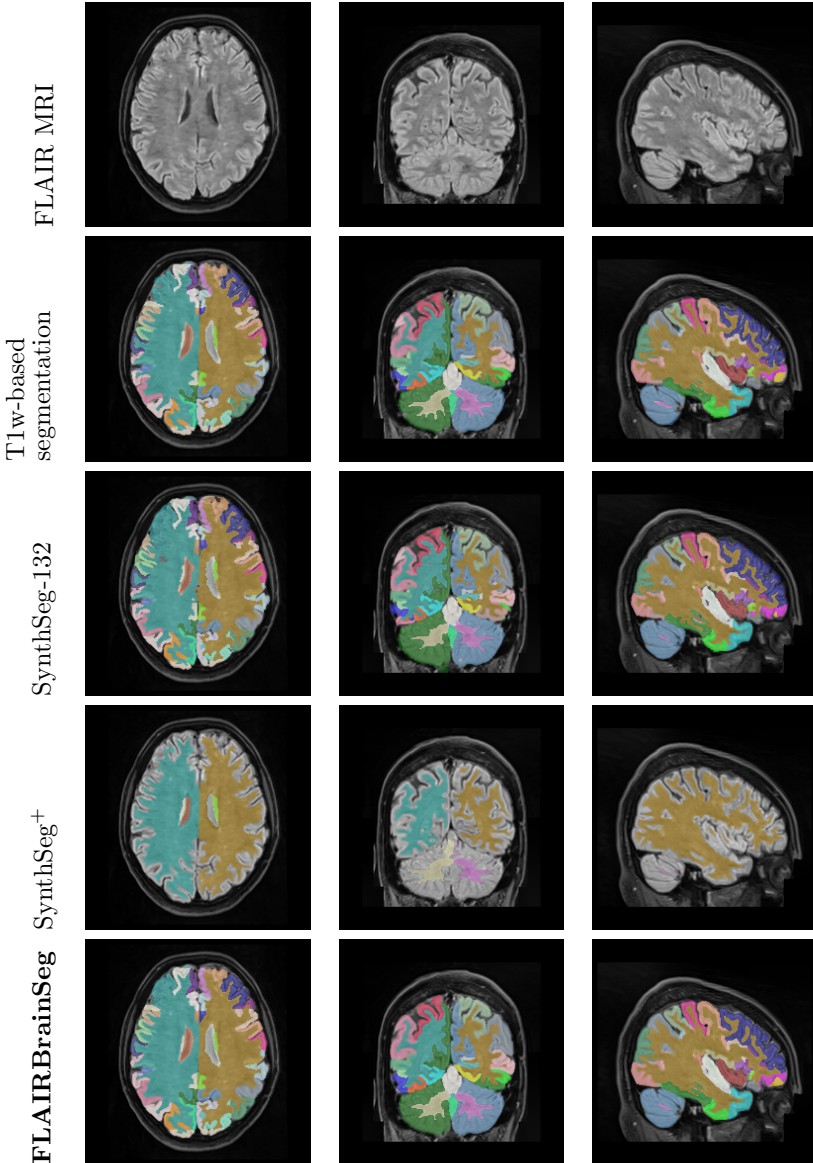

Figure 5: Examples of obtained segmentations with FLAIRBrainSeg, SynthSeg-132, and SynthSeg$^+$. Please note that the SynthSeg$^+$ segmentation includes only the 35 selected structures for comparison which do not include cortical structures and cerebellum grey matter.

## Appendix B. Average DSC by structure groups comparison

Fig. 6 illustrates the performance comparison in Experiment 1 between SynthSeg-132 and FLAIRBrainSeg across structure groups (e.g., tissue, cortical, subcortical, etc.) for the 132 structures defined in the Neuromorphometrics protocol on the in-domain test set.

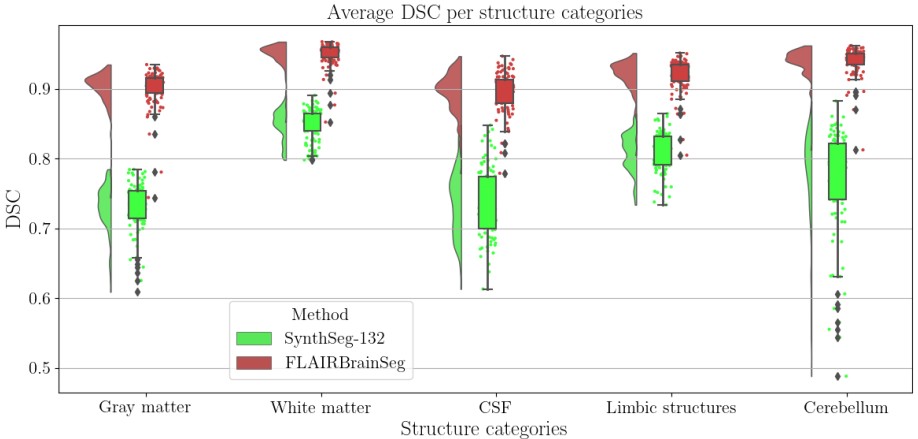

Figure 6: Comparison on Experiment 1 of SynthSeg-132 and FLAIRBrainSeg across structure groups (e.g., tissue, cortical, subcortical, etc.) for the 132 structures in the Neuromorphometrics protocol on the in-domain test set.

