# OpenReview forum: "FLAIRBrainSeg: Fine-Grained Brain Segmentation Using FLAIR MRI Only"
_MIDL.io/2025/Conference — MIDL 2025 Oral_

### Official Review · Reviewer_4uA3 · 2025-02-15

**Confidence:** 5
**Preliminary Rating:** 4
**Recommendation:** Oral
**Final Rating:** 5

**Summary:**

The paper introduces FLAIRBrainSeg, a novel method for fine-grained brain segmentation using only FLAIR MRI, in settings where T1w would not be acquired. The authors train a 3D UNet from the nn-Unet package on a dataset of ~3000 multiple-sclerosis patients. Labels are obtained by segmenting a T1 and coregistering FLAIRs and T1ws. MS lesions are inpainted in the T1w before the labelling to prevents mislabels and make the method robust. The authors perform quality-control on the training and testing data. The method is compared to two versions of SynthSeg, their method being more accurate.

**Strengths:**

This paper exhibits significant strengths in both data quality control and rigorous evaluation.  Several methods were employed to ensure the method received high-quality data as input, such as using another neural network to verify alignment between the T1w and FLAIRs, synthetic FLAIR generation to assess mislabels and more.

The authors retrained SynthSeg to fit their labelling scheme and managed to compare their method to SynthSeg+ [sic] even if the labels are not the same. The method exhibits strong performance in-distribution and generalizing to a different population.

**Weaknesses:**

While the method and its performance are certainly interesting, it may stem from a somewhat arbitrary problem formulation. Indeed, AFAIK T1w images are very rarely not acquired and T1w has been used extensively for brain segmentation. Moreover, page 2 mentions "This can also be an issue in existing public datasets, where T1w images are sometimes absent." No citations or example are provided which makes the reader wonder if there even is such a case. In this absence, the motivation behind this work are lessened.

Additonally, the lack of an open-source implementation and public distribution of reference labels and/or pretrained models is dissapointing. The conclusion mentions a wish for clinical relevance, which an open-source implementation would certainly help.

**Detailed Comments:**

Some more minor comments:

- Page 2 "In a second work" it would be a good idea to introduce the name "SynthSeg+" here as it is not what the original authors call it (SynthSeg 2.0 see README https://github.com/BBillot/SynthSeg) and Page 5 refers to "SynthSeg+" as if the reader was familiar with it.
- Section 2.4 does not need to be bullet points
- Section 2.4: how many subjects were excluded ? Does Table 1 list the subjects before or after QC ?
- Section 2.5: Why was the patch-size selected to be 112 × 128 × 112 ? It seems oddly specific. What is the size of the volumes ?
- Section 2.5 "Unlike the default nnU-Net configuration, augmentations did not include the flipping of images": why not ? is it due to labels being latteralized ?
-

**Justification Of The Final Rating:**

The authors have addressed the two major issues raised in my initial review: lack of citations for their key motivation and lack of open-source implementation. The authors have also addressed all of the minor comments. The quality of the final manuscript is certainly improved.

**Justification Of The Preliminary Rating:**

As mentionned above, while the key motivation behind this work may be on shaky grounds, the paper is overall very well writtent, the methodology is certain sound, the authors were thorough in their data labelling process and went to great lengths to provide a fair comparison with existing methods. This is overall a very good paper which is perfectly suited for MIDL.

**Questions To Address In The Rebuttal:**

Please address the comments wrt public datasets without T1ws. Most of the motivation behind the paper hinges on this. Please also address the minor comments mentionned above.

**Special Issue:**

No

---

> ### Author Response · Authors · 2025-03-07
> **Response to reviewer 4uA3**
>
> We thank reviewer 4uA3 for his review and valuable suggestions.
>
> **Weaknesses**
>
> *"The lack of an open-source implementation and public distribution of reference labels and/or pretrained models is disappointing."*
>
> We plan to publicly release an improved version of FLAIRBrainSeg. While we cannot commit to a specific timeline yet, the method will be freely made available in the future on the www.volbrain.net platform. We added this information in section 4.
>
> ---
>
> *"AFAIK T1w images are very rarely not acquired and T1w has been used extensively for brain segmentation."*
>
> There are specific clinical scenarios where T1w images are not routinely acquired or may not be available. To clarify, we have updated the introduction to include references to these clinical situations, particularly in multiple sclerosis (MS) and stroke, where the acquisition of T1w images is described as optional. Additionally, we have referenced Goodkin et al. (2021), which states that T1w sequences are "generally not part of the clinical MRI protocol." Given these points, the current understanding is that the acquisition of T1w images is considered optional in certain clinical settings, and if used, it is primarly for the application of brain volumetry or lesion segmentation. However, if FLAIR sequences enable accurate brain volumetry, T1w imaging may be omitted entirely.
>
>
> **Detailed Comments**
>
> *"In a second work" it would be a good idea to introduce the name "SynthSeg+" here as it is not what the original authors call it*
>
> We have updated section 1 to introduce the name of the method.
> The SynthSeg+ notation was adopted from the original authors PNAS paper (https://www.pnas.org/doi/epub/10.1073/pnas.2216399120).
>
> ---
>
> *Section 2.4 does not need to be bullet points*
>
> As per the reviewer's suggestion, we have updated Section 2.4 to remove the bullet points.
>
> ---
>
> *"Section 2.4: How many subjects were excluded? Does Table 1 list the subjects before or after QC?"*
>
> 1. For the OFSEP dataset, from the original 5065 OFSEP FLAIR images: 4891 images remained after the RegQCNet QC, 3577 images remained after the assessment of the correlation scores and the removal of 113 images visually assessed for the test set.
> 2. For the UKB dataset, the 100 test images were selected from the original 2954 images.
>
> To clarify this point, we added information on the number of visually assessed images in section 2.4.
>
> ---
>
> *"Section 2.5: Why was the patch size selected to be 112 × 128 × 112? It seems oddly specific. What is the size of the volumes?"*
>
> After resampling to the 1mm MNI space, the resolution of all processed volumes was identical and of size 181x217x181. The rules for selecting the patch size in nnU-Net are to use a patch "as big as possible" with regards to the VRAM available on the GPU, for training with a batch size of 2. The patch size was generated in the nnU-Net configuration step, when we used a Quadro RTX 8000 Nvidia GPU for training with 48GB of VRAM. We did not conduct experiments to compare performance using a different patch size.
>
> ---
>
> *Section 2.5 "Unlike the default nnU-Net configuration, augmentations did not include the flipping of images": why not? Is it due to labels being lateralized?*
>
> The reviewer was correct in assuming that flipping was excluded due to label lateralization. We have updated section 2.5 of the paper to include this information.

---

> > ### Comment · Reviewer_4uA3 · 2025-03-13
> >
> > The aurhors have done a tremendous job of addressing issues and concerned raised during my initial review and will adjust the final rating accordingly. I have no further comments.

---

### Official Review · Reviewer_vjGC · 2025-02-20

**Confidence:** 4
**Preliminary Rating:** 5
**Recommendation:** Poster

**Summary:**

This paper presents FLAIRBrainSeg, a deep-learning framework for fine-grained, whole brain segmentation. Using a training dataset with both T1w and FLAIR images for each subject, they produced ground truth FLAIR labels by first segmenting the T1w data using previously established methods (DeepLesionBrain and AssemblyNet). They then trained a 3D U-Net derived from nnU-Net to segment the FLAIR images. The authors validated their method against SynthSeg and SynthSeg+, and showed that FLAIRBrainSeg outperformed both SynthSeg variants in the in-domain and out-of-domain experiments. They also showed that their method was robust against MS lesions.

**Strengths:**

1. FLAIRBrainSeg appears to be a valuable contribution because it is able to produce fine-grained segmentations (132 labels) using only FLAIR images. This is advantageous in instances where T1w images may not be available, which are the most widely used modality in segmentation frameworks.
2. The quality control performed to ensure correspondence between the FLAIR images and ground truth labels generated from T1w images is quite thorough.
3. The paper is easy to read and well-written.

**Weaknesses:**

1. My main critique of this work is just structuring of the introduction section. I think it would be better to start the paper by presenting the most relevant issue, which is the need for segmentation frameworks specific to FLAIR.
2. What is the resolution of the FLAIR images compared to the T1w? I see that the data comes from a large number of different sites/scanners, so you could provide a range of resolutions if they are not all the same.
3. I think FLAIR images often have a very coarse slice resolution in clinical settings... would this impact the performance of your model? If so, then it may decrease its clinical applications. It would be helpful to discuss this is towards the end.

**Detailed Comments:**

In the introduction: The paper is about a segmentation framework for FLAIR images, but the first section focuses only on T1w data. Currently, it reads like this:
- whole brain segmentation with T1w MRI
- Modality-agnostic segmentation
- Need for non-T1w frameworks specific to other modalities
- Presenting FLAIRBrainSeg
I would restructuring this slightly by adding a few intro sentences at the very beginning to hook your readers with the main issue you are addressing. That would look more like this:
- Problem statement (need for FLAIR-specific segmentation)
- Most is T1w (review of this work)
- Some is more generalizable (review of modality-agnostic work)
- Presenting FLAIRBrainSeg
I would also present more specific examples of when we would only have FLAIR images to highlight why you chose FLAIR over another modality like proton density.
Also, if the authors find they need more room, Figure 3 is quite large. It could probably fit into a single line instead of a 2x2 grid.

**Justification Of The Preliminary Rating:**

The presented method is clinically significant and outperforms the relevant methods selected for comparison. The weaknesses I mentioned are minor and easily addressable. I also thought their method for quality control of T1w/FLAIR registration was quite clever (although this wasn't the main point of the paper).

**Questions To Address In The Rebuttal:**

Why did the authors select FLAIR as the specific modality?
What is the FLAIR resolution compared to T1w?
Can the model handle FLAIR images with more coarse resolutions?

---

> ### Author Response · Authors · 2025-03-07
> **Response to reviewer vjGC**
>
> We thank reviewer vjGC for his review and his valuable suggestion on the introduction. We have updated section 1 to clarify the problem statement and improving the introduction on the basis of the comments made by reviewer vjGC. We will try to address the weaknesses highlighted by the reviewer.
>
> **Weaknesses and Questions to Address in Rebuttal:**
>
> *"What is the resolution of the FLAIR images compared to the T1w?"*
>
> T1w and FLAIR images selected from the OFSEP database had a wide range of resolutions, ranging from 0.25x0.25x0.5mm to 2.0x2.0x3.0mm. ~50% of the available data had a 1mm isotropic sampling. We added this information in the paper (section 2.1) In most cases, T1w and FLAIR images had similar resolutions. To account for low-resolution images, the training data augmentations included a "low-resolution simulation" by resampling training images to lower resolution (up to 4mm).
>
> ---
>
> *"FLAIR images often have a very coarse slice resolution in clinical settings... would this impact the performance of your model?"*
>
> After the quality control steps, most of the selected images in the training/testing set were high-resolution (1mm or lower). We did not perform a more thorough analysis to assess the performance on lower-resolution images. The reviewer is right to inquire about the resolutions of training images, as it is currently a clear limitation of our work. Clinical FLAIR images often come at low resolution or as 2D images, with very few samples on the z-axis. We have updated the paper (section 4) to reflect this.
>
> ---
>
> *"Why did the authors select FLAIR as the specific modality?"*
>
> For transparency with reviewer vjGC, the motivation behind this work originated directly from our partners at hospitals specialized in MS and small vessel diseases. They highlighted having large amounts of unlabeled/unpaired FLAIR images and the need for processing methods tailored to these cases. This work benefited from the availability of a substantial number of FLAIR-T1w image pairs, which we did not have for other modalities, such as PD. We believe a similar pipeline could potentially be applied to other modalities, though some challenges specific to these modalities—particularly in the pre-processing steps—may arise.

---

> > ### Comment · Reviewer_vjGC · 2025-03-10
> >
> > Thank you for incorporating my suggestions into the final manuscript. I like the way that the authors decided to reformat the introduction section, and I have no further comments on the paper.
> >
> > One last thing--I would advise the authors to avoid addressing the anonymous reviewers with "he" and instead use a gender neutral "they", in case the reviewer happens to be a woman.

---

### Official Review · Reviewer_9NHh · 2025-02-22

**Confidence:** 3
**Preliminary Rating:** 4
**Recommendation:** Poster
**Final Rating:** 5

**Summary:**

This study presents FLAIRBrainSeg, a deep-learning-based pipeline for fine-grained brain segmentation using only FLAIR MRI scans. The results show that this model significantly outperformed the SynthSeg and SynthSeg+, in both in-domain and out-of-domain datasets.

**Strengths:**

The Dice score improvement from the baseline is significant, demonstrating the effectiveness of the proposed approach.
The authors provided detailed methodologies for each step in the pipeline, such as MS lesion inpainting and a thorough quality control process, which contribute to the robustness and accuracy of the segmentation.

**Weaknesses:**

The pipeline is well-structured, I appreciate the authors' work on this topic, which has made a significant contribution to the community. However, due to the complexity of the image processing pipeline, providing an analysis that quantifies the contribution of each component to the overall performance improvement would help readers better understand the paper. While several deep learning models are used in the pipeline, it is somewhat challenging to discern how each component progressively contributes to the development of this paper. For instance, how much of the performance improvement is attributable to QC?

**Detailed Comments:**

* 2.4. Quality control: What actions were taken with the subjects in the cluster of failed registrations?

* QC was applied to address the segmentation misalignment in 2.4, why not apply QC in the MS lesion segmentation (the MS lesion was segmented from Flair and then applied to T1w)?

**Justification Of The Final Rating:**

During my initial review of the manuscript, I had several concerns regarding the methodology and the results. In the rebuttal, the authors have successfully addressed my concerns and provided satisfactory explanations and revisions.

**Justification Of The Preliminary Rating:**

The work is well-designed, but a more detailed explanation of how each step in the pipeline contributes to the overall improvements would further strengthen the paper and enhance its clarity for readers.

**Questions To Address In The Rebuttal:**

See Weaknesses and detailed comments.

**Special Issue:**

No

---

> ### Author Response · Authors · 2025-03-07
> **Response to reviewer 9NHh**
>
> We thank reviewer 9NHh for his review. We will try to address the weaknesses highlighted by the reviewer to the best of our abilities.
>
> **Weaknesses**
>
> *"How much of the performance improvement is attributable to QC?"*
>
> We did not conduct experiments to measure the impact of QC on the Average DSC of FLAIRBrainSeg. We expect that training the model with misaligned images and segmentations would lead to a significant drop in performance.
>
> **Detailed Comments**
>
> *"What actions were taken with the subjects in the cluster of failed registrations?"*
>
> OFSEP training images where the post-QC registration error scores did not pass the selected threshold were removed from the training set entirely.
>
> ---
>
> *"Why not apply QC in the MS lesion segmentation?"*
>
> The registration between the T1w and the FLAIR image is performed only once. Misalignment observed using RegQCNet, or the proposed post-QC registration error, would also mean a misalignment between the MS WMH segmentation and the underlying WMH intensities. visual assessment of the MS Segmentation was not possible due to the large number of images used in this study. We did not introduce a specific QC method to assess the accuracy of the MS WMH segmentation.

---

### Official Review · Reviewer_hhDw · 2025-02-23

**Confidence:** 5
**Preliminary Rating:** 4
**Recommendation:** Oral, Poster
**Final Rating:** 5

**Summary:**

This paper introduces FLAIRBrainSeg, a novel deep learning-based method for fine-grained segmentation of brain structures using only FLAIR MRI scans. This is in contrast to most existing methods, which rely on T1-weighted (T1w) images for detailed anatomical segmentation. The authors train a 3D nnU-Net model to predict segmentations of 132 structures, typically derived from T1w images using AssemblyNet, directly from FLAIR images. They compare their method against two versions of SynthSeg: (1) a retrained SynthSeg (SynthSeg-132) using the same 132-structure protocol as FLAIRBrainSeg, and (2) SynthSeg+ (trained using multiple networs), using a subset of 35 common structures for comparison. Testing is performed on both an in-domain (OFSEP, n=100) and an out-of-domain (UK Biobank, n=100) dataset.

Results:

1. Superior Performance: For 2 experiments involving segmentation of 132-structures and 35-structures respectively, FLAIRBrainSeg significantly outperforms both SynthSeg-132  and SynthSeg+ on both in-domain and out-of-domain datasets, measured by Dice Similarity Coefficient (DSC) and 95th percentile surface distance (SD95).

2. Robustness to Pathology: white matter hyperintensities (WMH) associated with MS lesions are correctly classified and more accurately defined by FLAIRBrainSeg in the output segmentation compared to SynthSeg.

3. Computational Efficiency: FLAIRBrainSeg, using a single nnU-Net model, is faster at inference than SynthSeg+ and SynthSeg-132.

4. Out-of-Domain Generalization: The method generalizes well to the out-of-domain UK Biobank dataset which consists of healthy controls belonging to a different age group compared to the training data.

**Strengths:**

• The core idea of achieving fine-grained segmentation without T1w images is highly novel and clinically significant. It addresses a crucial limitation in scenarios where T1w scans are unavailable.


• The paper is well-written and organized. The pre-processing steps and the label-generation methodology applied by using the registration + segmentation pipeline involving AssemblyNet + DeepLesionBrain are described clearly.


• The authors use a well-established and openly available DL architecture (nnU-Net) and a large, multi-site training dataset (OFSEP). The experiments comparing FLAIRBrainSeg with SynthSeg are well-defined and carefully executed which also includes retraining SynthSeg on 132 structures to ensure a fair evaluation with the proposed method.


• The use of both in-domain and out-of-domain testing (using healthy controls from the UKBioBank) strengthens the experiments and evaluates the robustness of the method to out of domain data.


• Preliminary results shown in this paper demonstrate that a direct, supervised learning approach, as opposed to domain randomization (Synth-Seg), proves highly effective for fine-grained segmentation using only FLAIR data. The results are reported thoroughly citing relevant metrics such as the average dice score and SDM, and these are supplemented with relevant figures.

**Weaknesses:**

• Usage of the term "Bronze Standard": The target segmentations are derived from T1w images and are referred to as "bronze standard"  which is still an approximation of the ground truth. The term "bronze standard" occurs only a few times throughout the paper and can be potentially confusing. I would recommend removing the term altogether.


• The comparison with SynthSeg+ requires mapping between different segmentation protocols (Freesurfer and Neuromorphometrics). The authors acknowledge that this mapping is imperfect, introducing a potential confound. However, given the constraints, this is a reasonable approach.


•  The training data is exclusively from patients with multiple sclerosis (with no details on the data distribution such as stage of MS), and the out of domain testing is done on healthy controls. This raises the question of how well the method will generalize to other neurological conditions with different patterns of brain abnormalities (e.g., Alzheimer's disease, brain tumors, stroke, traumatic brain injury). The paper mentions extending the method to other pathologies in future work, but this is a key limitation in the current study.


For more details, please see the section "Questions to Address in the Rebuttal"

**Detailed Comments:**

The authors present a novel and clinically relevant method that achieves impressive results. The rigorous methodology, clear presentation, and strong performance, particularly in the presence of pathology, make this a valuable contribution. While some limitations exist, they are well-acknowledged and do not detract significantly from the overall impact of the work. This work has high potential for clinical translation and could significantly improve the analysis of FLAIR MRI data in various neurological applications.

**Justification Of The Final Rating:**

The post-rebuttal version of the paper submitted by the authors addresses the major points raised by the reviewers. This is a well-written, experimentally strong paper with important clinical implications, and highly relevant for the medical image analysis community.

**Justification Of The Preliminary Rating:**

This is a well-written, clinically relevant paper that proposes a novel pipeline to address the critical problem of segmentation of FLAIR datasets with important implications for clinical translation. The methodology and experimental design are thorough and well-structured. There are a few limitations in the testing and evaluation methods (which the author acknowledge) though they appear to be something that can be addressed in future work. If the rebuttal by the authors is satisfactory, I may consider increasing my rating.

**Questions To Address In The Rebuttal:**

• The performance of the downstream segmentation task relies on the accuracy of the preprocessing steps, particularly the lesion segmentation, inpainting and registration. The authors mention about incorporating Quality Control to ensure high accuracy in these steps and prevent any misalignment. Given that errors in these steps could propagate to the final segmentation, would you please clarify the range of errors obtained post QC, and the extent to which they affected the subsequent segmentation steps (i,e, did a lower post-QC registration or segmentation error score led to better average DSC or no major changes at all).

--------------
• In the Introduction section, correct “Consequently, we proposed a novel fine-grained….” to “Consequently, we propose a novel…”

--------------

• Table 2 showing the average DSC scores and SD95 are given in Appendix B. Given that one additional page can be added after acceptance, I would strongly recommend moving it to the main paper. In the Results section, I would recommend mentioning relevant scores and metrics wherever possible - terms such as "outperforms by a substantial margin" and "better accuracy" are ambiguous and these can be rephrased to "outperforms by a substantial margin ( 10% higher average DSC)".

-------------------

• The paper doesn't explicitly state the loss function, but nnU-Net typically uses a combination of Dice and Cross-Entropy. Was this used in this work ?

------------------
• Testing set: For the data selected for training and in-domain testing, were patients at a certain stage of MS selected or was this a random mixture of patients in different stages ? If the latter then what was the distribution, and how was this reflected in the in-domain testing stage. This information would be helpful to assess if there was no imbalance in the dataset towards a particular stage of MS.

-----------------

• Fig. 3: In-domain testing set; experiments 1 and 2, the readers would be interested in knowing more about the couple of “outlier” data points in the flair brain seg (which correspond to a lower dsc of 0.75, 0.8) - any insights into why they fall out of the range ?

-----------------

• It is mentioned that data augmentation techniques were applied to improve the training process. It would be helpful for the readers to know get more details on the different types of data augmentation used on the training set (a couple of sentences at max).

-----------------

--- **Minor questions and suggestions** ----


• In the AssemblyNet + DeepLesionBrain step, did the authors experiment replacing the AssemblyNet with the nn-Unet as well (or any other transformer architecture like Swin-unetr)

---------------------------------------

 • Train and evaluate without the initial denoising step and see how this would impact the overall performance.

----------------------------------------

• For future expansion into a possible journal paper, the authors can consider a set of ablation studies involving different augmentation methods, pre-processing pipelines (with and without Assembly Net + DeepLesionBrain).

-----------------------------------

**Special Issue:**

Yes

---

> ### Author Response · Authors · 2025-03-07
> **Response to reviewer hhDw**
>
> We thank reviewer hhDw for his very thorough review. We will, when possible, address the weaknesses highlighted by the reviewer, as well as answer the reviewer questions to the best of our abilities.
>
> **Weaknesses**
>
> *"The term "bronze standard" occurs only a few times throughout the paper and can be potentially confusing."*
>
> We introduced the term to convey and highlight the fact that segmentations used in this work are made from T1w images, and are thus an approximation of the manual segmentation (considered the gold standard). We agree that it might be confusing and replaced all instances of "bronze standard" with "T1w-based segmentation."
>
> **Questions to Address in Rebuttal:**
>
> *"Please clarify the range of errors obtained post-QC, and the extent to which they affected the subsequent segmentation steps."*
>
> To address the reviewer’s question, we assessed the range of post-QC registration error scores for the OFSEP test images in relation to the Average DSC. The post-QC registration error scores ranged from 0.4 to 0.71.
>
> While investigating the outliers in the distribution of Average DSC, we noticed that the low post-QC registration error score corresponded to the image with the lowest Average DSC. This observation demonstrates that the QC process, when applied, is effective in identifying misalignments, as low post-QC registration error scores were associated with misaligned images.
>
> However, it is important to note that a high post-QC registration error score does not necessarily imply a high Average DSC, indicating that while the QC process helped identify misalignments, it does not guarantee segmentation accuracy.
>
> ---
>
> *"The readers would be interested in knowing more about the couple of “outlier” data points in the FLAIRBrainSeg."*
>
> Please see the answer above.
>
> ---
>
> *WRT reviewer suggestion on the paper (introduction correction, appendix table results, results description)*
>
> We followed the reviewer’s suggestion and updated section 3 of the paper to match the reviewer’s recommendation.
>
> ---
>
> *"nnU-Net typically uses a combination of Dice and Cross-Entropy. Was this used in this work?"*
>
> The loss function used in this work is indeed the combination of DiceLoss and CrossEntropy introduced in nnU-Net. We have not explored alternative loss functions in this work. We have updated section 2.5 of the paper to include this information.
>
> ---
>
> *"Were patients at a certain stage of MS selected or was this a random mixture of patients in different stages?"*
>
> For clarification in the repartition of patients of the in-domain dataset test set and training set: The stratified train-test split was performed with regards to the patients' age group and sex only. The information on the EDSS was available to us (for some, not all) of the OFSEP dataset.
>
> The EDSS distribution in the training set ranged from 0 to 9.0 (with a median value of 2.0). The EDSS distribution in the test set ranged from 0 to 6.0 (with a median value of 1.5). We have updated section 2.1 to include this information in the paper.
>
> Following the reviewer’s comments, to explore the potential relationship between EDSS and segmentation accuracy, we performed a linear regression analysis between EDSS and Average DSC for cases where EDSS was available. The results showed no significant association between the two variables (p = 0.468, R² = 0.014), suggesting that there is no strong relationship between segmentation accuracy and EDSS.
>
> ---
>
> *"More details on the different types of data augmentation used on the training set."*
>
> Data augmentations included in this work are the same data augmentations introduced in nnU-Net. More specifically, they can be broadly divided into three parts:
>
> 1. Linear Spatial Augmentations (to be precise, this includes rotations and scaling)
>  2. Intensity Augmentations (Gaussian Noise, Gaussian Blur, Brightness, Contrast, etc...)
>  3. Low-resolution resampling (up to 4mm) We have updated section 2.3 of the paper to include this information.
>
>
> **Minor Questions and Suggestions**
>
> *"Did the authors experiment replacing the AssemblyNet with the nn-Unet as well?"*
>
> We did not make any experiments to compare the performance of different methods for the segmentation of the OFSEP T1w.
>
> ---
>
> *"Train and evaluate without the initial denoising step and see how this would impact the overall performance."*
>
> We did not make any experiments with regards to the removal of the denoising steps. We agree with the reviewer that it would be interesting.
>
> ---
>
> *"The authors can consider a set of ablation studies involving different augmentation methods, pre-processing pipelines (with and without Assembly Net + DeepLesionBrain)."*
>
> We thank the reviewer and completely agree with their suggestion.

---

> > ### Comment · Reviewer_hhDw · 2025-03-11
> >
> > Thank you for incorporating the reviewers' suggestions into the final manuscript. The authors have addressed the major points raised by the reviewers and the updated version is good to go. I have changed my rating to 'Strong Accept', and I have no further comments on the paper.

---

### Official Review · Reviewer_vnwu · 2025-03-02

**Confidence:** 5
**Preliminary Rating:** 5
**Recommendation:** Best Paper Award, Oral

**Summary:**

The authors introduce a framework, FLAIRBrainSeg, for segmenting fine anatomical structures in the brain using FLAIR images. Often, this segmentation task is performed on T1-weighted images. The FLAIRBrainSeg method can segment the brain even in the absence of the preferred data format, i.e., T1-weighted images.

The authors have utilized T1-weighted MRI images to construct 'bronze standard' segmentation maps. Lesion segmentation was performed on the T1-weighted MRI, and later, those lesions were inpainted so that the fine anatomical structure segmentation focuses on brain regions and parts rather than being affected by lesions. The segmentation maps obtained from the T1-weighted MRI were thus focused on anatomical structures and were used as labels for the FLAIR images.

The authors chose AssemblyNet for performing this fine anatomical structure segmentation on the T1-weighted MRI. Since the labels generated in this process were to be used for segmenting the same structures on FLAIR images, the authors aligned the T1-weighted MRI images to ensure that the labels matched the position and orientation on the FLAIR images. To remove segmentation misalignment, a FLAIR-like image was generated using the local median intensity values of each region corresponding to a structure label. This image, referred to as the synthetic FLAIR image, was used to reduce misalignment between the generated labels and the FLAIR images.

A correlation value was computed and then fitted with a two-component Gaussian Mixture Model (GMM)—one component for failed registrations and another for successful registrations. A threshold was determined to filter out improper registrations. The images that were successfully registered, as indicated by the correlation value, were visually assessed for possible misalignment.

The remaining FLAIR segmentations were considered as good labels for downstream FLAIR image segmentation tasks involving fine anatomical structures. The authors used nnU-Net for this segmentation task. They compared their method with two other methods—SynthSeg and SynthSeg+. The testing approach required customization, as the settings of SynthSeg and SynthSeg+ differed from those of FLAIRBrainSeg. FLAIRBrainSeg outperformed both SynthSeg and SynthSeg+ in fine anatomical structure segmentation on both in-domain and out-of-domain datasets.

**Strengths:**

The authors have proposed a novel framework, FLAIRBrainSeg, for fine anatomical structure segmentation. The idea and process of creating the bronze segmentation map using T1-weighted MRI are interesting. The framework is novel, and the authors have clearly outlined the steps for building it by diligently describing each component—dataset preparation, bronze map construction, mitigation of alignment issues, fine anatomical structure segmentation, and customization of baseline methods for comparison. The paper is clear and easy to understand.

**Weaknesses:**

The testing strategy had a lot of customization, which might not be easy to reproduce though the framework building process was easy to follow and should be reproducible.

I did not understand how the FLAIR like image was created using the local median intensity value.

**Detailed Comments:**

The paper adds a novel framework and performs better in the task of fine anatomical structure. The paper reads well and conveys the idea clearly. Thanks to the authors for marking the circle on Figure 4. top right image

Some details regarding the visual assessment are lacking . For example
1.  who did the visual assessment?
2. How many iages were visually assessed (page 4 last line)
3. How many images remained after the visuall assessment?
4. How many were filtered out?

**Justification Of The Preliminary Rating:**

This is a novel framework targeting fine anatomical structure segmentation from a specific modality like FLAIR. This framework comprises of multiple innovative blocks  such as bronze map creation, visual assessment and Gaussian Mixture Model fitting for filtering misaligned scans and a customized testing strategy. In my opinion this paper has a significant contribution which will be of interest to the community.

**Questions To Address In The Rebuttal:**

The authors can share details about  how the FLAIR like image was created using the local median intensity value as that would strengthen the manuscript.

The authors can add the following details in this paper
1. Who did the visual assessment?
2. How many images were visually assessed (page 4 last line)
3. How many images remained after the visual assessment?
4. How many were filtered out?

---

> ### Author Response · Authors · 2025-03-07
> **Response to reviewer vnwu**
>
> We thank reviewer vnwu for his very encouraging review. We will, when possible, address the weaknesses highlighted by the reviewer, as well as answer the reviewer questions to the best of our abilities.
>
> **Weaknesses**
>
> *"I did not understand how the FLAIR-like image was created using the local median intensity value."*
>
> The synthetic FLAIR MRI was generated as follows, using both the FLAIR and T1w-based segmentation:
> 1. For every segmentation label, we used the intensities of the FLAIR image underneath the obtained segmentation to compute the median of each structure.
> 2. We created a new image, where, for each segmentation label, in place of the segmentation, we put the corresponding median intensity observed above. We used the median intensity to ensure robustness to "outlier" intensity values that might come from other structures due to small misalignments.
> 3. To render the image more realistic, we used a uniform filter to simulate the partial volume effect.
>
> We have updated the description of the synthetic FLAIR generation process in the paper (section 2.4)
>
> **Detailed comments**
>
> *"Some details regarding the visual assessment are lacking."*
>
> To answer the reviewer’s questions:
>
> The visual assessment was performed by the authors themselves, and not a clinician. When performing the assessment, the authors looked for >1mm/1vox visible displacement, mainly on cortical structures.
>
> We assessed all the test images (thus, 213 images were visually assessed this way) for the constitution of the test set. To clarify, in order to construct the test set, we performed:
>
> 1. A stratified train-test split of all available images before the application of QC based on the sex and age groups of the patients.
>
> 2. A visual assessment of the test images. In cases where the quality of an image was not satisfactory, the image was removed entirely from the list of available images, and a new train-test split to select replacement test images was performed. 100 images from the UKB dataset for the constitution of the test set were visually assessed. 113 images from the OFSEP dataset for the constitution of the test set were visually assessed. 13 images were removed during the assessment. We have updated the manuscript (section 2.4) to describe this process and address the reviewer's comments.

---

> > ### Comment · Reviewer_vnwu · 2025-03-14
> >
> > Thank you for the additional information. I do not have any more questions.

---

### Author Rebuttal · Authors · 2025-03-07

**Rebuttal:**

We would like to sincerely thank the reviewers for their time, insightful comments, and valuable feedback, which have significantly contributed to improving the quality of this work. We have updated the manuscript to account for the proposed improvements and have highlighted the changes made from the current version. Additionally, we have provided individual responses to each of the five reviews we've received.

**Supporting Material:**

/attachment/f662e1beda38356f959d4407381a2d1a4bed70f0.pdf

---

### Meta-Review · Area_Chair_Btju · 2025-03-20

**Recommendation:** Accept (Oral)
**Confidence:** 5

**Metareview:**

All reviewers found the proposed method to be novel and the results promising. The paper introduces FLAIRBrainSeg, a deep-learning framework for fine anatomical brain segmentation using only FLAIR MRI. This approach is particularly valuable in clinical settings where T1w scans are unavailable. The method outperforms SynthSeg and SynthSeg+ on both in-domain and out-of-domain datasets, demonstrating strong generalization and clinical utility.

The authors provided a well-structured methodology, incorporating T1w-based segmentation labels, MS lesion inpainting, and a GMM-based quality control pipeline. The quality control process was rigorous, ensuring that misaligned scans were filtered out. Fair comparisons were conducted by retraining SynthSeg on the same segmentation protocol, further strengthening the evaluation.

Concerns raised by the reviewers, including the use of the term "bronze standard," lack of citations on T1w absence, and the need for open-source implementation, were adequately addressed in the rebuttal. The authors clarified terminology, provided additional references, and committed to publicly releasing the model. Minor issues related to dataset resolution variability and segmentation accuracy were acknowledged as future research directions.